# Task-Driven Out-of-Distribution Detection with Statistical Guarantees for Robot Learning

**Alec Farid**[*]  **Sushant Veer**[*]  **Anirudha Majumdar**

Department of Mechanical and Aerospace Engineering, Princeton University

{afarid, sveer, ani.majumdar}@princeton.edu

**Abstract:** Our goal is to perform *out-of-distribution (OOD) detection*, i.e., to detect when a robot is operating in environments that are drawn from a different distribution than the environments used to train the robot. We leverage Probably Approximately Correct (PAC)-Bayes theory in order to train a policy with a *guaranteed bound* on performance on the training distribution. Our key idea for OOD detection then relies on the following intuition: violation of the performance bound on test environments provides evidence that the robot is operating OOD. We formalize this via statistical techniques based on p-values and concentration inequalities. The resulting approach (i) provides guaranteed confidence bounds on OOD detection, and (ii) is *task-driven* and sensitive only to changes that impact the robot's performance. We demonstrate our approach on a simulated example of grasping objects with unfamiliar poses or shapes. We also present both simulation and hardware experiments for a drone performing vision-based obstacle avoidance in unfamiliar environments (including wind disturbances and different obstacle densities). Our examples demonstrate that we can perform task-driven OOD detection within just a handful of trials. Comparisons with baselines also demonstrate the advantages of our approach in terms of providing statistical guarantees and being insensitive to task-irrelevant distribution shifts.

**Keywords:** Out-of-distribution detection, generalization, PAC-Bayes

## 1 Introduction

Imagine a drone trained to perform vision-based navigation using a dataset of indoor environments and deployed in environments with varying wind conditions, obstacle densities, or lighting (Fig. 1). Similarly, consider a robot arm manipulating a new set of objects or an autonomous vehicle deployed in a new city. State-of-the-art techniques for learning-based control of robots typically struggle to generalize to such *out-of-distribution* (OOD) environments. This lack of OOD generalization is particularly pressing in safety-critical settings, where the price of failure is high. In this work, we focus on the problem of autonomously *detecting* when a robot is operating in environments drawn from a different distribution than the one used to train the robot. This ability to perform *OOD detection* has the potential to improve the safety of robotic systems operating in OOD environments. For example, a drone operating in a new set of environments could either deploy a highly conservative policy or cease its operations altogether. In addition, OOD detection can also allow the robot to improve its policy by re-training using additional data collected from the new environments.

There are two important desiderata that OOD detection approaches for safety-critical robotic systems should ideally satisfy. First, we would like to develop OOD detection techniques with *guaranteed confidence bounds*. Second, we would like our OOD detectors to be *task-driven* and only sensitive to *task-relevant* changes in the robot's environment. As an example, consider again the drone navigation setting in Fig. 1 and suppose that the robot's policy is insensitive to changes in color and lighting. Here, the robot's OOD detector should *not* trigger even if the robot is operating in environments with different color/lighting and should only trigger if there are task-relevant variations (e.g., variations in the obstacle density). Unfortunately, current approaches (Sec. 2) do not typically satisfy both desiderata; they are often based on heuristics and not task-driven in general.

*Statement of Contributions.* We develop task-driven OOD detection techniques with statistical guarantees on correctness. To this end, we make three specific contributions (see Fig. 1 for an overview).

---

[*]Equal Contribution

5th Conference on Robot Learning (CoRL 2021), London, UK.

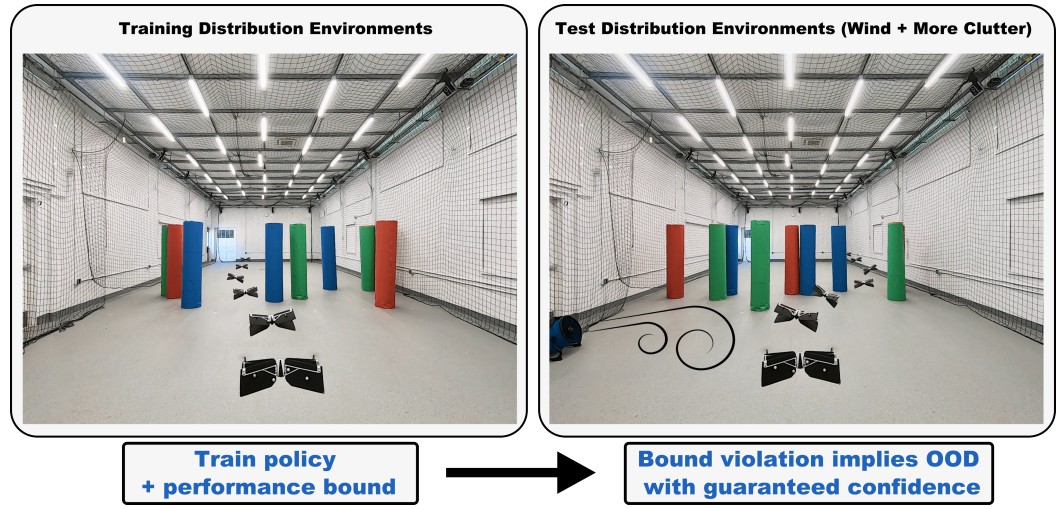

Figure 1: A schematic of our overall approach. We learn policies with guaranteed bounds on expected performance on the training distribution. Violation of this bound during deployment implies that the robot is operating OOD (with high confidence). We present simulation experiments in manipulation and navigation settings, and hardware experiments for a drone navigating in new environments with varying wind conditions and clutter.

- Given a dataset of environments drawn from an (unknown) training distribution, we develop a pipeline based on *generalization theory* for training control policies with a *guaranteed bound* on performance (a bound on the expected cost of the policy on the unknown training distribution). Specifically, we leverage recently-developed *derandomized Probably Approximately Correct (PAC)-Bayes* bounds that are well-suited to enable OOD detection (Sec. 4.1).

- We develop two OOD detection techniques with complementary statistical interpretations based on the following intuition (Sec. 4.2): if the costs incurred when the robot is deployed in new environments violate the bound on the policy's performance, then this indicates that the robot is operating OOD. We formalize this via p-values and concentration inequalities and provide statistical guarantees on the false positive rate of OOD detection. Since our OOD detection scheme leverages the costs incurred in new environments, it is *only* triggered by task-relevant changes.

- We demonstrate our approach on two simulated examples (Sec. 5): (i) a robotic manipulator grasping a new set of objects, and (ii) a drone navigating a new set of environments. Comparisons with baselines demonstrate the advantages of our approach in terms of providing statistical guarantees and being insensitive to task-irrelevant shifts. We also present a thorough set of hardware experiments for vision-based drone navigation with varying wind conditions and clutter (Fig. 1). Our experiments demonstrate the ability of our approach to perform task-driven OOD detection within just a handful of trials for systems with complex dynamics and rich sensing modalities.

## 2 Related work

**Anomaly/OOD detection in supervised learning.** Anomaly detection in low-dimensional signals has been well-studied in the signal processing literature (see [1] for a review). Recent work in machine learning has focused on OOD detection for high-dimensional inputs (e.g., images) in supervised learning settings (see [2] for a review). Popular approaches use threshold-based detectors for the output distribution of a given pre-trained neural classifier [3, 4, 5]. Other methods use a specific pipeline for training a neural network in order to improve OOD detection on test samples [6, 7, 8]. However, these methods are often susceptible to adversarial attacks [9]. Thus, approaches for addressing adversarial data have been developed [10, 9, 11]. Some of these approaches are also able to provide theoretical guarantees of performance on adversarial data [9, 12, 13]. Other methods provide PAC-style statistical guarantees [14, 15] or p-values [16]. However, these methods typically focus on supervised learning settings and often require specific network outputs (e.g., softmax) that are incompatible with non-classification tasks. In contrast, we focus on OOD detection for policy learning settings in robotics and do not make assumptions about the specific structure of the policy.

**Task-driven OOD detection.** The methods above are aimed at detecting *any* distributional shift in the data and can be sensitive even to task-*irrelevant* shifts (i.e., ones that do not impact performance) as we demonstrate in our experiments (Sec. 5). A recent approach determines an estimate of input

atypicality for pre-trained networks and uses it as an OOD detector in supervised learning settings [17]. Recent methods have also been developed specifically for reinforcement learning (RL) [18, 19, 20, 21, 22]. In particular, [22] presents a general task-driven approach for OOD detection on sequential rewards, which is optimal in certain settings. However, neither this method nor others in the RL context provide statistical guarantees on detection. We propose an OOD detection framework which is both task-driven and provides statistical guarantees by leveraging generalization theory.

**Generalization theory.** Generalization theory provides a way to learn hypotheses (in supervised learning) with a bound on the true expected loss on the underlying data-generating distribution given only a finite number of training examples. Original frameworks include Vapnik-Chervonenkis (VC) theory [23] and Rademacher complexity [24]. However, these methods often provide vacuous generalization bounds for high-dimensional hypothesis spaces (e.g., neural networks). Bounds based on PAC-Bayes generalization theory [25, 26, 27] have recently been shown to provide strong guarantees in a variety of settings [28, 29, 30, 31, 32, 33], and have been significantly extended and improved [34, 35, 36, 37, 38, 39]. PAC-Bayes has also recently been extended to learn policies for robots with guarantees on generalization to novel environments [40, 41, 42, 43]. In the present work, we leverage recently-proposed *derandomized* PAC-Bayes bounds [44]; this framework allows us to train a single deterministic policy with a guaranteed bound on expected performance on the training distribution (in contrast to [40, 41, 42, 43], which train stochastic neural network policies). This forms the basis for our OOD detection framework: by observing violations of the PAC-Bayes bound on test environments, we are able to perform task-driven OOD detection with statistical guarantees.

## 3   Problem formulation

**Dynamics and environments.** Let $s_{t+1} = f_E(s_t, a_t)$ describe the robot's dynamics, where $s_t \in \mathcal{S} \subseteq \mathbb{R}^{n_s}$ is the state of the robot at time-step $t$, $a_t \in \mathcal{A} \subseteq \mathbb{R}^{n_a}$ is the action, and $E \in \mathcal{E}$ is the environment that the robot is operating in. "Environment" here broadly refers to factors that are external to the robot, e.g., a cluttered room that a drone is navigating, disturbances such as wind gusts, or an object that a manipulator is grasping. The dynamics of the robot may be nonlinear/hybrid. We denote the robot's sensor observations (e.g., RGB-D images) by $o_t \in \mathcal{O} \subseteq \mathbb{R}^{n_o}$.

**Cost functions.** The robot's task is encoded via a cost function and we let $C_E(\pi)$ denote the cost incurred by a (deterministic) policy $\pi$ when deployed in environment $E$ over a finite time horizon $T$. The policy $\pi \in \Pi$ is a mapping from (histories of) sensor observations to actions (e.g., parameterized using a neural network). In the context of obstacle avoidance, the cost could capture how close the drone gets to an obstacle; in the context of grasping, the cost could be 0 if the robot successfully lifts the object or 1 otherwise. We assume that the cost is bounded; without further loss of generality, we assume $C_E(\pi) \in [0, 1]$. We also assume that the robot has access to the cost $C_E(\pi)$ after performing a rollout on $E$ (i.e., at the end of an episode of length $T$). This is a relatively benign assumption in robotics contexts since the cost function often has physical meaning and can be measured by the robot's sensors. For example, a drone equipped with a depth sensor can measure the smallest reported depth value during its operation in an environment, and a manipulator equipped with a camera and/or force-torque sensor can measure if it successfully grasped an object. We make no further assumptions on the cost function (e.g., we *do not* assume continuity, Lipschitzness, etc.).

**Training and testing distribution.** We assume that the robot has access to a training dataset $S = \{E_1, \ldots, E_m\}$ of $m$ environments drawn i.i.d. from a training distribution $\mathcal{D}$, i.e. $S \sim \mathcal{D}^m$. After training, the robot is deployed on environments in $S' = \{E_1', \ldots, E_n'\}$ drawn from a test distribution $\mathcal{D}'$: $S' \sim \mathcal{D}'^n$. Importantly, we *do not* assume any explicit knowledge of $\mathcal{D}, \mathcal{D}'$ or the space $\mathcal{E}$ of environments. We only have indirect access to $\mathcal{D}, \mathcal{D}'$ in the form of the finite training datasets $S, S'$.

**Goal: task-driven OOD detection.** After being deployed in (a typically small number of) environments in $S'$, the robot's goal is to detect if these environments were drawn from a different distribution than the training distribution (i.e., if $\mathcal{D}'$ is different from $\mathcal{D}$). Moreover, our goal is to perform *task-driven* OOD detection. In particular, our OOD detector should only trigger if:

$$C_{\mathcal{D}'}(\pi) \coloneqq \mathop{\mathbb{E}}_{E' \sim \mathcal{D}'} C_{E'}(\pi) > C_{\mathcal{D}}(\pi) \coloneqq \mathop{\mathbb{E}}_{E \sim \mathcal{D}} C_E(\pi) \ . \tag{1}$$

Thus, our OOD detector should be *insensitive* to changes in the environment distribution that do not adversely[1] impact the robot's performance. This is a challenging task since we only assume

---

[1]The OOD detection scheme we present can also be modified to detect if the expected cost on $\mathcal{D}'$ is *smaller* than the expected cost on $\mathcal{D}$; however, we focus on the other case since that is of greater practical interest.

access to a finite number of environments from $\mathcal{D}$ and $\mathcal{D}'$. Moreover, our goal is to develop an OOD detection framework that is broadly applicable in challenging settings involving nonlinear/hybrid dynamics, rich sensing modalities (e.g., RGB-D), and neural network-based policies.

## 4 Approach

Our overall approach is illustrated in Fig. 1. First, we train a policy with an associated *guarantee* on the expected cost on the training distribution $\mathcal{D}$ (Sec. 4.1). We then apply our OOD detection scheme which formalizes the following intuition: violation of the bound during deployment implies (with high confidence) that the test distribution $\mathcal{D}'$ is OOD in a task-relevant manner (Sec. 4.2).

### 4.1 Policy training via derandomized PAC-Bayes bounds

Given a training dataset $S = \{E_1, \ldots, E_m\}$ of $m$ environments drawn i.i.d. from the training distribution $\mathcal{D}$, our goal is to learn a policy $\pi$ with a guaranteed bound on the expected cost $C_{\mathcal{D}}(\pi) := \mathbb{E}_{E \sim \mathcal{D}} C_E(\pi)$. Since our OOD detection scheme will rely on violations of the bound, it is important to obtain bounds that are as tight as possible. In this work, we utilize the *Probably Approximately Correct (PAC)-Bayes* framework [25, 26, 27] to train policies with strong guarantees. More specifically, we leverage recently-developed *derandomized* PAC-Bayes bounds [44], which are well-suited to the OOD detection setting (as we explain further below).

PAC-Bayes applies to settings where one chooses a *distribution* over policies (e.g., a distribution over weights of a neural network), and learning algorithms that have the following structure: (1) choose a *"prior"* distribution $P_0$ over the policy space $\Pi$ *before* observing any data (this can be used to encode domain/expert knowledge); (2) obtain a training dataset $S$ and choose a *posterior* distribution $P$ over the policy space $\Pi$. Let $P$ be the output of an algorithm $A$ which takes $P_0$ and $S$ as input. Denote the cost incurred by a policy $\pi$ on the training environments in $S$ as $C_S(\pi) := \frac{1}{m} \sum_{E \in S} C_E(\pi)$. The following result is our primary theoretical tool for training policies with bounds on performance.

**Theorem 1** *For any distribution $\mathcal{D}$, prior distribution $P_0$, $\delta \in (0, 1)$, cost bounded in $[0, 1]$, $m \geq 8$, and deterministic algorithm $A$ which outputs the posterior distribution $P$, we have the following:*

$$\mathbb{P}_{(S,\pi) \sim (\mathcal{D}^m \times P)} \left[ C_{\mathcal{D}}(\pi) \leq \overline{C}_{\delta}(\pi, S) \right] \geq 1 - \delta \ , \tag{2}$$

*where $\overline{C}_{\delta}(\pi, S) := C_S(\pi) + \sqrt{R}$, $R := \left( D_2(P \| P_0) + \ln \frac{2\sqrt{m}}{(\delta/2)^3} \right) / (2m)$, and $D_2$ is the Rényi Divergence for $\alpha = 2$ defined as: $D_2(P \| P_0) = \ln \left( \mathbb{E}_{\pi \sim P_0} \left[ \left( \frac{P(\pi)}{P_0(\pi)} \right)^2 \right] \right)$.*

*Proof.* The proof is in App. A.1. We use [44, Theorem 2], a general pointwise PAC-Bayes bound. We perform the reduction from supervised learning to policy learning presented in [40]. □

This result allows us to obtain policies with guaranteed bounds on the expected cost. In particular, we can search for a posterior $P$ in order to minimize the sum of the training cost and the "regularizer" $\sqrt{R}$. We describe such training methods via backpropagation and blackbox optimization below. Sampling from the resulting posterior $P$ provides a policy with a bound on $C_{\mathcal{D}}(\pi)$ that holds with high probability (over the sampling of the training dataset $S$ and the policy $\pi$).

Recent work has demonstrated the effectiveness of PAC-Bayes to provide strong bounds for deep neural networks [28, 31, 33] and specifically for policy learning [40, 41, 42, 43]. However, the bounds used by these approaches do not provide a viable approach for performing OOD detection. The approaches are based on traditional PAC-Bayes bounds, where a *distribution* $P$ over policies (e.g., a distribution over neural network weights) is chosen; the resulting bound is on $\mathbb{E}_{\pi \sim P} C_{\mathcal{D}}(\pi)$ instead of $C_{\mathcal{D}}(\pi)$. Thus, given a test dataset $S'$ of environments, many policies from the distribution $P$ must be sampled in order to evaluate/bound the expected cost on $S'$. This is not feasible in an OOD detection setting, where there is single execution on the test environments. Our use of the derandomized PAC-Bayes bound in Theorem 1 avoids this issue since we can bound $C_{\mathcal{D}}(\pi)$ for a *particular* policy sampled from $P$.

We provide approaches for optimizing the bound provided in Theorem 1 using backpropagation (App. A.4) and Evolutionary Strategies (ES) [45] (App. A.5). Since Theorem 1 requires a deterministic training algorithm, we fix the random seed for stochastic training methods. This makes the algorithm deterministic as the same input will always produce the same output. We choose multivariate Guassian distributions with diagonal covariance $\text{diag}(s)$, i.e., $P = \mathcal{N}(\mu, \text{diag}(s))$, for the

posterior $P$ and prior $P_0$ distributions. Further, let $\psi := (\mu, \log s)$; we use the shorthand $\mathcal{N}_\psi$ for $\mathcal{N}(\mu, \mathrm{diag}(s))$. We denote $\pi_w$ with weights $w \sim \mathcal{N}_\psi$ as a parameterization of the robot's policy (e.g., neural networks with weights $w$). After training, we sample and fix a $w$ from the trained posterior for deployment on test environments. We then compute the PAC-Bayes upper bound $\overline{C}_\delta(\pi, S)$.

## 4.2 Task-driven OOD detection with statistical guarantees

We now tackle the problem of OOD detection as defined in Sec. 3. The PAC-Bayes training pipeline from Sec. 4.1 produces a policy $\pi$ and an associated bound $\overline{C}_\delta(\pi, S)$ on the expected cost $C_{\mathcal{D}}(\pi)$ that holds with probability $1 - \delta$ over the sampling of the training dataset $S \sim \mathcal{D}^m$ and the policy $\pi \sim P$. Our key idea for OOD detection is that if our PAC bound $\overline{C}_\delta(\pi, S)$ is violated by $\pi$ in the test environments $S'$ (drawn from the test distribution $\mathcal{D}'$), then this indicates that the test environments are OOD. We formalize this intuition below using two popular frequentist statistical inference tools — hypothesis testing via p-value and confidence interval overlap.

**Method 1: Hypothesis testing**

In accordance with our notion of task-driven OOD detection (1) described in Sec. 3, let $H_0$ be the null-hypothesis which claims that $C_{\mathcal{D}'}(\pi) \leq C_{\mathcal{D}}(\pi)$ and let $H_1$ be the alternate hypothesis which claims that $C_{\mathcal{D}'}(\pi) > C_{\mathcal{D}}(\pi)$ for a given policy $\pi$. Then, to perform a hypothesis test, we compute the p-value and check if it drops below a significance level $\alpha \in (0, 1)$, which is chosen before looking at the data. The p-value bounds the probability of drawing test environments with average cost larger than the observed test cost $C_{S'}(\pi)$, assuming that the null-hypothesis holds. In the event that the p-value is smaller than $\alpha$, we can conclude that under the null-hypothesis the observed test dataset $S'$ had a very small probability of being drawn; therefore, the null-hypothesis $H_0$ can be rejected. A mathematically precise definition of the p-value is given as follows.

**Definition 1** (adapted from [46]) *Let $\mathcal{D}'$ be the test distribution and $S' \sim \mathcal{D}'^n$ be an observed dataset. Let $\pi$ be the robot's control policy. Then, the p-value is defined as:*

$$p(S') := \mathbb{P}_{\hat{S} \sim \mathcal{D}'^n} [C_{\hat{S}}(\pi) \geq C_{S'}(\pi) \mid H_0] \ . \tag{3}$$

Note that $\hat{S}$ is *any* dataset of cardinality $n$ drawn i.i.d. from the test distribution $\mathcal{D}'^n$, while $S'$ is the observed test dataset which we must use for inference. We present an upper bound on the p-value by leveraging the PAC-Bayes generalization bound (Theorem 1). This upper bound holds with probability $1 - \delta$ (over the sampling of $S$ and $\pi$).

**Theorem 2** *Let $\mathcal{D}$ be the training distribution and $P$ be the posterior distribution on the space of policies obtained through the training procedure described in Sec. 4.1. Let $S' \sim \mathcal{D}'^n$ be a test dataset, $p(S')$ be the p-value for $S'$ defined in Definition 1, and $\delta \in (0, 1)$. Then,*

$$\mathbb{P}_{(S, \pi) \sim (\mathcal{D}^m \times P)} [p(S') \leq \exp(-2n\overline{\tau}(S)^2)] \geq 1 - \delta \ , \tag{4}$$

*where $\overline{\tau}(S) := \max\{C_{S'}(\pi) - \overline{C}_\delta(\pi, S), 0\}$.*

*Proof.* The proof relies on Hoeffding's inequality and is provided in App. A.2. □

Theorem 2 provides an upper bound on the p-value which holds with high confidence. If the upper bound is below the significance level $\alpha$, then with high confidence we can say that the p-value is below $\alpha$; thereby, Theorem 2 facilitates OOD detection through hypothesis testing.

**Method 2: Confidence interval on the difference in expected train and test costs**

We now present another method for performing task-driven OOD detection (Sec. 3 and Eq. (1)) by lower bounding $C_{\mathcal{D}'}(\pi) - C_{\mathcal{D}}(\pi)$ with high probability. If the lower bound is positive, then with high confidence we can conclude that the robot is operating in (task-relevant) OOD environments as the test cost is larger than the train cost. Furthermore, this lower bound serves as a measure of how "far" the two distributions are from the perspective of the task.

**Theorem 3** *Let $\mathcal{D}$ be the training distribution, $\mathcal{D}'$ be the test distribution, and $P$ be the posterior distribution on the space of policies obtained through the training procedure described in Sec. 4.1. Let $\delta, \delta' \in (0, 1)$ such that $\delta + \delta' < 1$, $\gamma := \sqrt{\frac{\ln(1/\delta')}{2n}}$, and $\Delta C := C_{S'}(\pi) - \gamma - \overline{C}_\delta(\pi, S)$. Then,*

$$\mathbb{P}_{(S, \pi, S') \sim (\mathcal{D}^m \times P \times \mathcal{D}'^n)} [C_{\mathcal{D}'}(\pi) - C_{\mathcal{D}}(\pi) \geq \Delta C] \geq 1 - \delta - \delta' \ . \tag{5}$$

*Proof.* A detailed proof of this theorem is provided in App. A.3. □

To use Theorem 3 for detecting if a test dataset $S'$ is OOD, we first choose $\delta$ and $\delta'$ to obtain a desired confidence level $1 - \delta - \delta'$ for our OOD detector. Using these constants and the policy $\pi$ drawn from $P$ trained using the PAC-Bayes upper bound, we then compute $\Delta C$. If $\Delta C > 0$, then with confidence at least $1 - \delta - \delta'$ we can claim that $C_{\mathcal{D}'} > C_{\mathcal{D}}$, and therefore, with the same level of confidence the test dataset $S'$ must be OOD according to our task-relevant notion in (1). Furthermore, we obtain a guaranteed upper bound of $\delta + \delta'$ on the false positive rate of our OOD detector, i.e., the rate at which our detector will misclassify training distribution data as OOD.

## 5  Examples

We demonstrate the ability of our approach to perform task-driven OOD detection with guaranteed confidence bounds on two simulated examples: a manipulator grasping a new set of objects and a drone navigating a new set of environments. For the navigation task, we compare our methods with popular OOD detection baselines. Our code is available at: https://github.com/irom-lab/Task_Relevant_OOD_Detection

### 5.1  Robotic grasping

**Overview.** We use the Franka Panda arm (Fig. 2(a)) for grasping objects in the PyBullet simulator [47] and build upon the open-source code provided in [42]. The robot employs a vision-based control policy that uses a depth map of the object obtained from an overhead camera and returns an open-loop action $a := (x, y, z, \theta)$ which corresponds to the desired grasp position and yaw orientation of the gripper. We train the manipulator to grasp mugs placed in $SE(2)$ poses drawn from a specific distribution. Then, we demonstrate the efficacy of our OOD detection framework by (i) gradually modifying the distribution on the mug poses and (ii) changing the objects from mugs to bowls.

**Control policy.** The control policy is a deep neural network (DNN) which inputs a $128 \times 128$ depth map of the object and a latent state $z \in \mathbb{R}^{10}$ sampled from a multivariate Gaussian distribution $\mathcal{N}_\psi$ with a diagonal covariance, and outputs an open-loop grasp action $a$; see Fig. 4 in App. A.6.1 for the policy. In [42], the distribution $\mathcal{N}_\psi$ on the latent space encodes prior domain/expert knowledge.

**Training.** Mugs from the ShapeNet dataset [48] are randomly scaled in all dimensions to generate a training dataset $S$ of 500 mugs. If the robot is able to lift the mug by 10 cm and the gripper palm does not contact it, then we consider the rollout successful and assign a cost of 0; otherwise the cost is set to 1. In training, we optimize the distribution $\mathcal{N}_\psi$ on the latent space to minimize the PAC-Bayes upper bound provided in Theorem 1 using Alg. 2, while the weights of the CNN and MLP networks in Fig. 4 in App. A.6.1 remain fixed. The prior $\mathcal{N}_{\psi_0}$ is chosen as the normal distribution with zero mean and identity covariance. A policy $\pi$ is sampled from the trained posterior $\mathcal{N}_\psi$ and the PAC-Bayes bound for this policy is computed as $\overline{C}_\delta(\pi, S) = 0.1$ with $\delta = 0.01$.

**OOD detection results.** We perform OOD detection using the two methods presented in Theorem 2 and Theorem 3. For detection with p-value, we choose a significance level $\alpha = 95\%$, while, for detection using $\Delta C$ we choose a confidence level of 95%, i.e., $\delta + \delta' = 0.05$, which ensures that the false-positive rate of our detector is no greater than 5%. We perform two experiments to demonstrate the efficacy of our approach for OOD detection. First, we make the distribution on the mug's initial placement progressively more challenging; see App. A.6.1 for the exact distributions. For each distribution, we sample 20 test datasets of cardinality 10 and compute our OOD indicators: (i) the lower bound on $1 - p$ (where $p$ is the p-value) and (ii) $\Delta C$ using Theorem 3. Fig. 2(b) plots the mean (dashed line) and a one standard deviation spread (shaded region) for the OOD indicators as a function of $C_{\mathcal{D}'} - C_{\mathcal{D}}$ (estimated via exhaustive sampling). Note that we plot $\Delta C + 0.95$ so that the OOD threshold is the same (0.95) for both methods. As the cost of the policy deteriorates on test distributions our OOD indicators reliably increase, capturing the shift of the test distributions away from the training distribution. In the second experiment, we change the objects that the manipulator must grasp from mugs to bowls. Fig. 2(c) shows that with a small test dataset $S'$ of cardinality 5, both our approaches detect OOD when bowls are used (red curves). As expected, our OOD detectors are not triggered for mugs (blue curves), which are drawn from the training distribution.

### 5.2  Vision-based obstacle avoidance with a drone

**Overview.** In both the simulation and hardware portions of this example, we aim to avoid an obstacle field with the Parrot Swing drone; this is an agile quadrotor/fixed-wing hybrid drone shown in Fig. 1.

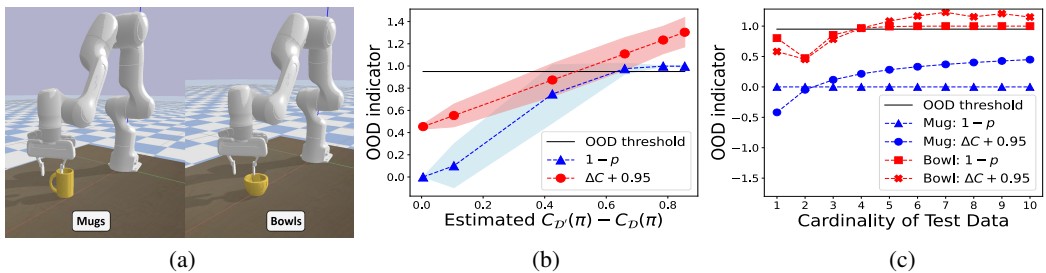

Figure 2: OOD detection for grasping. **(a)** Franka Panda arm in PyBullet grasping a mug (left) and a bowl (right). **(b)** Performance of our OOD detectors for different distributions on mug placement. Both our approaches perform similarly and the OOD indicators increase monotonically with $C_{\mathcal{D}'}(\pi) - C_{\mathcal{D}}(\pi)$. **(c)** Comparison of our OOD detectors for grasping mugs and grasping bowls. Both our approaches detect OOD using a small number of test environments (just 4) for bowls and do not detect OOD for mugs (as expected).

We train a DNN control policy in a simulation setup based on the hardware system shown in Fig. 1. The policy takes in a $50 \times 50$ depth image and outputs a softmax corresponding to a set of pre-computed motion primitives with the goal of avoiding obstacles by the largest distance. Since we designed the simulation portion of this example with application to hardware in mind, we have created motion primitives by capturing (with a Vicon motion tracking system) the trajectories of open-loop control inputs, which result in different maneuvers; the two images in Fig. 1 represent two of these trajectories. The use of motion primitives allows us to perform accurate sim-to-real transfer (since the motion primitives are recorded directly from the hardware system).

**Training.** Environments consist of a set of randomly placed cylindrical obstacles. We record the minimum distance $d_{\min}$ from the obstacles (as recorded by the robot's $120°$ field of view depth sensor) and assign a cost of $\max(0, 1 - \frac{d_{\min}}{30\text{cm}})$. Using $10{,}000$ training environments $S$, we train a prior to assign larger values to motion primitives which achieve a larger distance from obstacles. See App. A.6.2 for further details on the training procedure for the prior. We then use another $10{,}000$ environments to train the posterior distribution using Alg. 1. We sample a policy $\pi$ from the trained posterior and compute the PAC-Bayes bound $\overline{C}_\delta(\pi, S) = 0.222$ for $\delta = 0.01$.

**Simulation Results**

We compare our task-driven OOD detection approach with two baselines: (i) maximum softmax probability (MSP) [3] (an effective and popular baseline for OOD detection), and (ii) MaxLogit [5] (a recent state-of-the-art OOD detection baseline). We note that these baselines are specifically designed for networks which output categorical distributions, and thus provide strong benchmarks. We determine the threshold for detection for the baselines by choosing a false positive rate of $5\%$ for the training environments. However, note that the baselines do not provide any guarantees; they may violate the false positive rate even on new environments drawn from the training distribution (as we observe in our experiments). We select a p-value of $0.05$ as the indication that an environment is OOD, and a guaranteed false-positive rate of $\delta + \delta' = 5\%$ for the confidence interval method. Each method receives a dataset $S'$ of 10 environments. We use the average outputs of the baselines on the 10 environments for OOD detection. We generate OOD environments of varying difficulty by changing the number of obstacles and the maximum or minimum gap-size between obstacles.

**Varied environment difficulty.** We sample 2000 datasets $S'$ (of size 10) for each setting and plot the proportion of these detected as OOD for all methods in Fig. 3(a). At 0 on the x-axis of this plot, we draw datasets from the training distribution. All other environments are OOD. We note that the p-value and confidence interval-based methods perform similarly. The plot demonstrates our ability to perform task-driven OOD detection. In particular, we only detect the "harder" distributions (i.e., environments with higher increased cost) as OOD, while the baselines are triggered even for the easier distributions. In addition, neither baseline provides any guarantees and both violate the $5\%$ false-positive rate on the training distribution. Our methods have a low false positive rate, albeit at the cost of conservatism for moderate distribution shifts.

**Task-irrelevant shift.** We also compare the baselines with our methods on a distribution where environments consist of 4 (uniformly) randomly located obstacles. In this setting, the control policy achieves a near-identical expected cost $C_{\mathcal{D}'}(\pi)$ (as estimated by exhaustive sampling of environments) to the expected training cost $C_{\mathcal{D}}(\pi)$ (in particular, $C_{\mathcal{D}'}(\pi)$ - $C_{\mathcal{D}}(\pi) = -0.02$). For this setting, MSP [3] classified $95.7\%$ of test datasets as OOD and MaxLogit [5] classified $50.9\%$ as

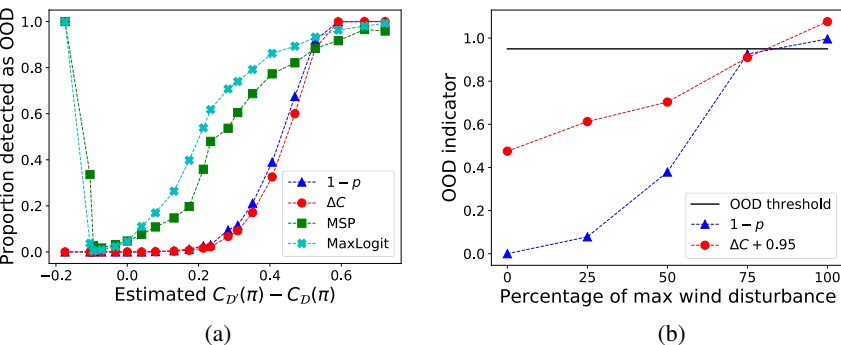

(a)                                                                    (b)

Figure 3: OOD Detection on settings of variable difficulty. **(a)** Comparison of the performance of our OOD detectors with baselines MSP [3] and MaxLogit [5]. Both our methods do not detect OOD when $C_{\mathcal{D}'}(\pi) < C_{\mathcal{D}}(\pi)$ whereas the baselines detect these task-irrelevant shifts in the environment. For $C_{\mathcal{D}'}(\pi) > C_{\mathcal{D}}(\pi)$ the baselines detect a higher proportion of datasets as OOD for smaller values of $C_{\mathcal{D}'}(\pi) - C_{\mathcal{D}}(\pi)$ compared to our methods. **(b)** Comparison of our OOD detectors on the Parrot Swing hardware for increasing wind disturbance. Both OOD indicators increase monotonically with $C_{\mathcal{D}'}(\pi) - C_{\mathcal{D}}(\pi)$ and are able to detect OOD at $100\%$ wind.

OOD. Thus, the baselines are triggered by a task-irrelevant shift in the distribution. In contrast, our methods had an OOD detection rate of $0.1\%$ in this setting.

**Hardware Results**

We use a Parrot Swing drone for the hardware experiments (Fig. 1). We simulate a depth sensor for the drone (as if the sensor was mounted on the drone) by generating a synthetic depth image using the positions of objects from the Vicon motion capture system. We do not provide any other information to the policy (e.g., obstacle positions). We generate each environment the same way as in simulation, and then place the real-world obstacles in the generated locations.

**Varied environment difficulty and wind disturbances.** We deploy the policy trained in simulation on three kinds of OOD environments in hardware: (i) environments with a smaller number of obstacles ("easier" environments), (ii) environments with smaller gaps between obstacles (i.e., "harder" environments), and (iii) environments with wind generated using a fan (Fig. 1 right) with the same obstacle distribution as training. Videos from each setting are available at https://youtu.be/VxKCAjaih8M. For each setting, we run 10 trials on the hardware and use this for OOD detection. As expected, our OOD detectors are not triggered by the easier environments. For the harder environments, we compute $1 - p \geq 0.81$ and $\Delta C = -0.11$. Results from the windy environments are shown in Fig. 3(b) for increasing values of wind (up to about $5$ m/s). We note that the sim-to-real distribution shift (corresponding to the zero wind case) is not viewed as being OOD in a task-relevant manner by our approaches. Both our approaches assign increasingly high confidence as the wind speed is increased. We note that disturbances such as wind cannot be detected via the depth image given to the robot's policy. Thus any OOD detection technique which relies solely on the output of the policy, such as MSP [3] and MaxLogit [5], would be unable to detect these environments as OOD.

## 6 Conclusion

We have presented a framework for performing task-driven OOD detection with statistical guarantees. Our approach uses PAC-Bayes theory to train a policy with a bound on the expected cost on the training distribution. We then perform OOD detection on test environments by checking for violations of the bound (using p-values and concentration inequalities). Our simulated and hardware experiments demonstrate the ability of our approach to perform OOD detection within a handful of trials. Comparisons with baselines also demonstrate two advantages: our OOD detectors (i) are only sensitive to task-relevant distribution shifts, and (ii) provide statistical guarantees on detection.

**Challenges and future work.** The approach we present here allows us to bound the false positive rate. An interesting theoretical question is whether we can also bound the false negative rate. It would also be of practical interest to extend our approach to settings where the robot encounters environments in an *online* manner (instead of the batch setting we consider here). Another particularly exciting direction is to develop versions of our approach that are more *proactive*; instead of having to incur costs on the test environments, one could potentially perform OOD detection based on *predicted* costs (thus avoiding the need to potentially fail on the test environments).

**Acknowledgments**

The authors were supported by the Office of Naval Research [N00014-21-1-2803, N00014-18-1-2873], the NSF CAREER award [2044149], and the Toyota Research Institute (TRI). This article solely reflects the opinions and conclusions of its authors and not ONR, NSF, TRI or any other Toyota entity.

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
