# OpenReview forum: "Task-Driven Out-of-Distribution Detection with Statistical Guarantees for Robot Learning"
_robot-learning.org/CoRL/2021/Conference — CoRL2021 Poster_

### Official Review · Reviewer_ev1V · 2021-07-05

**Originality:** Good
**Technical Quality:** Good
**Clarity Of Presentation:** Good
**Impact:** 2

**Recommendation:**

Weak Accept: I recommend accepting the paper, but will not argue for my recommendation if the majority of other reviewers have a different opinion.

**Summary:**

The paper presents a PAC-Bayes approach for out-of-distribution (OOD) detection within policy learning settings. The authors first apply derandomized PAC Bayes to train a policy, which provides a bound per hypothesis rather than averaged analysis over multiples hypothesis. Then, as a bound is provided per hypothesis, the paper proposes to check the violation of the PAC-Bayes guarantees at test time, i.e. if the PAC Bayes bound is violated, the authors attribute this to OOD scenario. Concretely, this is achieved via hypothesis testing and checking the confidence intervals. Theoretically, a bound is established for these two schemes, which holds within a pre-specified interval. The method is task driven, as the expected costs are used. Experiments with 2 simulations and 1 real robot - in particular, the drone performing vision-based obstacle avoidance - the authors validate the approach. Overall, the paper provides an interesting application of PAC-Bayes to OOD detection within the policy learning settings.

**Issues:**

- eq (4) §\hat{S}§ seems undefined.

- eq (2), eq (3), eq (5), eq (6): P is undefined (though standard in PAC-Bayes, may not be for OOD detection communities).

- figure 2 & 3: indicate what is better or worse, or what the data means intuitively.

- other comments on writing above.

- more comparisons to the state-of-the-art.

- A.17 to A.20 and A.27 to A.30 needs more explanations.

- figure 3 missing the plots of varying environmental difficulties and only shows wind conditions variations.

- setting m bigger than 8, how does this influence the derandomized PAC Bayes? At testing, do you need a batch as well?

- The paper trains a policy with derandomized PAC Bayes. Would this affect the performance of the trained policy over the previous PAC-Bayes policies OR even the state-of-the-art methods? How difficult is it to train as well? Also, how tight is the bound in practice?

- In the appendix, the authors refer to different sources. For the researchers working on OOD detection, providing more background materials in the appendix can help in understanding the paper. For the researchers familiar with PAC-Bayes methods, it can help in not resorting to other papers.

- An empirical validation of the proposed statistical guarantees can be meaningful, e.g. histogram analysis?


-----------------------------------------------------------------------------------------------------------------------------------------------

Post Rebuttal

-----------------------------------------------------------------------------------------------------------------------------------------------

I thank the authors for the detailed response. I have thoroughly read the comments from other reviewers, the author response and the revised manuscript. I am happy to see that the revised manuscript has improved in its clarity.

However, I maintain my score; I find the paper to be a borderline due to the following reasons:

-  (On significance) more informative comparison to the state-of-the-art.

The provided use-case of the method is clearly on OOD detection for the policy learning, and yet, the advancements to the OOD literature is unclear.

My first point is on the unclear motivation behind the chosen baselines. As evidenced by Section 2, the authors state that the missing gaps in "Anomaly/OOD detection in supervised learning" is the need for the specific network outputs, e.g. softmax (classification) and perhaps covariance (regression). However, the authors still compare to the approaches for supervised learning: MSP and MaxLogit. Therefore, I found the motivation behind the chosen baselines unclear.

My second point is the need to choose other "task-driven OOD detection approaches" as the baselines. As evidenced by Section 2, the authors discuss other "task-driven OOD detection" approaches, and point out that the missing gaps are the statistical guarantees. However, the advancements to the literature is unclear as the authors do not provide any comparison to other "task-driven OOD detection" approaches. Therefore, I found the advancements to the state-of-the-art unclear.

While the authors address my concerns to an extent, I still found the arguments not fully convincing. For example, the argument that MaxLogit (in 2017) has shown to outperform other alternatives is not convincing. This is because the settings of the experiments are different, and the comparisons of MaxLogit are limited to the approaches prior to 2017. Moreover, the reason behind not comparing to other "task-driven OOD detection" approaches is because the proposed method requires the "roll-out" in some test environments, which is the limitation of the proposed method over the existing methods. In this regard, the readers could appreciate more if the authors have resolved this limitation, and provided a comparison to the other task-driven OOD detection methods.

This is the reason why I rate the significance of the paper low; the paper doesnt have to outperform the other approaches, but should contain the required information for the readers to illustrate the advancements to the current approaches.

-  (On significance) the required roll-outs in test environments.

Taking more global view, many existing approaches do not require the roll-outs in the test environments, which is the main point for developing OOD detectors (targeting at safety-critical applications). Simple examples are the baselines chosen by the authors - MSP [3] and MaxLogit [5] (and many others, e.g. reference [4, 17]). Therefore, my question still remains: how meaningful it is to have the provided method with some guarantees, when the resulting failures are catastrophic (over the existing task-driven methods despite the given drawback of the approach).

-  (On thoroughness) the analysis of the batch-size at testing.

The proposed method can be sensitive to the chosen batch-size at testing (as the authors provide the information). Then, perhaps, an ablation study that guides the choice of an optimal batch-size would be meaningful.

-  (On thoroughness) more data samples for varying environment difficulty.

Only 2 evaluation points for varying environment difficulty might also be limited in thoroughness, as oppose to more data points with wind variations. The paper could improve with more data samples for varying environment difficulty.


-----------------------------------------------------------------------------------------------------------------------------------------------

Post Rebuttal

-----------------------------------------------------------------------------------------------------------------------------------------------

**Reviewer Expertise:**

Very good: Comprehensive knowledge of the area

**Strengths And Weaknesses:**



Pros:

- To my knowledge, the proposed application of PAC-Bayes to OOD detection within the policy learning settings is novel and interesting. Some of the derivation steps that the authors demonstrate, are non-trivial and thought involving.

- The paper's focus on "task-driven" OOD detection - only figuring out the OOD samples that affect the performance of the policy - is relevant to safety-critical settings of robot learning, and the paper shows real robot experiments as well as some of the theoretical foundations associated with their method. This is likely to be appreciated at CoRL.

Cons

- The paper has several rooms for improving the exposition.

First, the margins seem to be somehow reduced, when compare to other CoRL submissions. This can be checked and corrected if inappropriate, during the rebuttal period. Second, while the insights are very clear, it might help to provide some remarks at the end of each theorems - ideally, in a common tongue or insights. Third, for the proofs in the Appendix, it might help to first state the proof paths, and provide the proofs, so that the work is more accessible. For the video, you may want to make it standalone. Currently, it is difficult to get the relevant information before reading the paper. For example, we see the drone crash at OOD settings, but there are no explanations.There are also minor comments which I list at the end of the review.

- The comparisons to the existing approaches are kept minimal when compared to rational works on OOD detection. A compelling comparisons to other OOD methods can further show-case the strength of PAC-Bayes approaches within the considered scenario.

To back up, the paper compares to MSP - iclr 2017 and MaxLogit - arxiv 2019 (figure 3), which are both targeted for general settings, and it is unclear whether MSP and MaxLogit are the state-of-the-art alternatives. The paper should demonstrate how far the idea has advanced the state-of-the-art OOD detection literature, e.g. compared to other PAC-Bayes OOD detection methods (ref 14-16), as well as other task-driven OOD methods such as ref 18-22.

The reason for emphasizing the comparison to the state-of-the-art is that the provided bounds are statistical, and there are chances that the guarantees are violated. An interesting question is then, how meaningful it is to have such guarantees, when the resulting failures are catastrophic.

**Summary Of Recommendation:**

I recommend accepting this paper. The paper provides principled approach to OOD detection using PAC-Bayes bounds. It seems original, and technical quality is good - the approach is reasonably backed up by the existing theory, and the real robot experiments show-case the success of their method, which is likely to be appreciated at CoRL.

My main concern is on the clarity of presentation (which is good but still has rooms to improve - the bars are usually high with theory) and lack of rigorous comparisons to the state-of-the-art alternatives.

---

> ### Author Response · Authors · 2021-08-28
> **Response to Reviewer ev1V**
>
> Thank you for your thorough review and valuable suggestions. We address your comments and suggestions below.
>
> > the margins seem to be somehow reduced, when compare to other CoRL submissions
>
> We have fixed the layout and formatting of the paper. Additionally, to improve the flow of the paper we have moved some training-related technical details to the Appendix. We have also added further explanations and details as requested by reviewers (highlighted in blue in the revision and discussed here in our responses to reviewers).
>
> > while the insights are very clear, it might help to provide some remarks at the end of each theorems - ideally, in a common tongue or insights
>
> Thank you for this suggestion. We have included remarks at the end of Theorem 2 as well to clarify its contribution.
>
> > for the proofs in the Appendix, it might help to first state the proof paths, and provide the proofs, so that the work is more accessible
>
> We have included proof pathways in the beginning of the proofs provided in the Appendix. For your convenience, the pathways are presented below:
>
> Proof pathway for Theorem 2:
>
> We prove this theorem by considering two cases: when the PAC-Bayes cost inequality holds, i.e., $C_\mathcal{D}(\pi)$ $\leq \overline{C}_\delta(\pi,S)$, and when it does not, i.e., $C_\mathcal{D}(\pi)$ $> \overline{C}_\delta(\pi,S)$; the two cases are considered in (A.7)-(A.9). In the latter case, we cannot say anything about the p-value, while in the former case, which holds with probability at least $1-\delta$, we show in (A.10)-(A.21) that $p(S') \leq \exp(-2n\overline{\tau}(S)^2)$.
>
>
> Proof pathway for Theorem 3:
>
> To lower bound the difference between $C_{\mathcal{D}'}(\pi)$ and $C_{\mathcal{D}}(\pi)$ with high probability we obtain a lower bound on $C_{\mathcal{D}'}(\pi)$ which holds with probability at least $1-\delta'$ using Hoeffding's inequality in (A.23)-(A.26). Then we use this bound with the PAC-Bayes bound (2) which holds with probability at least $1-\delta$ to obtain (A.22) by following the steps in (A.27)-(A.32).
>
> > For the video, you may want to make it standalone. Currently, it is difficult to get the relevant information before reading the paper. For example, we see the drone crash at OOD settings, but there are no explanations.
>
> We are creating a new self-contained video which communicates the message of the paper and better explains the hardware results. To allow more time for discussion of our responses, we are replying to these comments before the video is finished and will update the submission with the video prior to the end of the discussion period.
>
> > The comparisons to the existing approaches are kept minimal when compared to rational works on OOD detection. A compelling comparisons to other OOD methods can further show-case the strength of PAC-Bayes approaches within the considered scenario.
> To back up, the paper compares to MSP - iclr 2017 and MaxLogit - arxiv 2019 (figure 3), which are both targeted for general settings, and it is unclear whether MSP and MaxLogit are the state-of-the-art alternatives. The paper should demonstrate how far the idea has advanced the state-of-the-art OOD detection literature, e.g. compared to other PAC-Bayes OOD detection methods (ref 14-16), as well as other task-driven OOD methods such as ref 18-22.
>
> We believe that MSP and in particular MaxLogit are effective and widely-used baselines to compare to. In [5], the authors test MaxLogit against strong baselines and show that it greatly outperforms many existing baselines. We also compare against MSP since it is widely popular and provides a standard comparison.
>
> We provide guarantees for task-relevant shifts (in the policy's cost) to the distribution from which test samples are drawn. The guarantees provided in [14-16] are on a somewhat different notion of OOD. These papers focus on guaranteed detection for any samples which do not come from the same distribution as the training data. Thus, we do not believe the two types of guarantees are easily reconciled or compared.
>
> A challenge with comparing against current task-driven OOD methods, which focus on reinforcement learning, is the requirement of an online reward. The problem of OOD detection within a rollout is important for safety-critical applications and work to address this will be part of future work (we expand upon this point below).
> With such a method, comparing against existing methods for OOD detection given an online reward would be informative.
> However, we do not believe there is a clear way to both give task-driven OOD methods for reinforcement learning an online reward and provide a fair comparison with our method, where the reward is only received at the end of a rollout.

---

> > ### Author Response · Authors · 2021-08-28
> > **Continued Response to Reviewer ev1V**
> >
> > > An interesting question is then, how meaningful it is to have such guarantees, when the resulting failures are catastrophic.
> >
> > Our approach does not necessarily require the system to fail; deterioration of the costs is sufficient to perform OOD detection. Indeed, this is demonstrated by our drone experiments where we are able to detect OOD environments without collisions since we rely on a cost signal that captures distance from the obstacles. Similarly, for the grasping example, one could perform OOD detection without dropping objects if one has access to a tactile sensor -- the cost signal would then correspond to the distance of the ratio of the normal and tangential forces from the boundary of the friction cone. Analogously, a self-driving car can use its distance to nearby obstacles/agents as a cost.
> >
> > However, we do agree with the reviewer that OOD detection without performing entire rollouts in test environments is a problem of great importance for safety-critical applications and we are currently working on addressing it. The following provides an insight into our approach for tackling this challenge:
> >
> > To achieve OOD detection with guaranteed false positive rate without performing entire rollouts, we are training a network that takes in a sensor measurement in a given environment and returns a prediction of the cost if the policy were to be deployed in this environment. We provide a PAC bound on the predicted cost; it is worth clarifying that the PAC bound is not on the true expected cost, but it is on the expected predicted cost for training environments. We leverage the insight that if the predicted cost for the test environments violate the PAC bound for the predicted cost on training environments, it indicates that the test environments are OOD. Finally, we note that we could perform this sort of OOD detection with a PAC bound on any scalar statistic; however, we use the predicted cost because it would allow us to embed task-relevance through the cost-predictor network. If a feature does not affect the cost of the policy, the cost predictor would ideally learn to discount it and focus on the task-relevant features.
> >
> > > eq (4) $\hat{S}$ seems undefined.
> >
> > $\hat{S}$ is any dataset of cardinality $n$ drawn i.i.d. from the test distribution $\mathcal{D}'^n$. On the other hand $S'$ is the observed test dataset which we must use for inference. We have included this clarification in lines 203-204.
> >
> > > eq (2), eq (3), eq (5), eq (6): P is undefined (though standard in PAC-Bayes, may not be for OOD detection communities).
> >
> > We have included a more complete introduction to the posterior distribution in lines 149-150.
> >
> > > figure 2 and 3: indicate what is better or worse, or what the data means intuitively.
> >
> > We have included the interpretation of the plots in the captions of Figures 2 and 3.
> >
> > > A.17 to A.20 and A.27 to A.30 needs more explanations.
> >
> > We have included more detailed explanations for the equations mentioned by the reviewer.
> >
> > > figure 3 missing the plots of varying environmental difficulties and only shows wind conditions variations.
> >
> > The hardware results for varying environment difficulty are described in the text on lines 331-333. Since there are only two data points, we did not include a plot.
> >
> > > Setting m bigger than 8, how does this influence the derandomized PAC Bayes?
> >
> > The cardinality of the training dataset is denoted by $m$. For $m\geq 8$, the derandomized PAC-Bayes bound presented in Theorem 1 holds. We have revised the statement of Theorem 1 to make this explicit. The larger the $m$, the tighter the PAC-Bayes bound is because the regularizer asymptotically converges to 0 as $m$ increases. For $m<8$ the result of Theorem 1 does not hold; however, this is not really a limitation because a training dataset with cardinality less than $8$ is highly unusual. We emphasize that the cardinality restriction is only on the training dataset used to obtain the PAC-Bayes bound and associated policy; the test dataset (using which OOD detection is performed) can have arbitrary cardinality (and indeed typically has small cardinality).
> >
> > > At testing, do you need a batch as well?
> >
> > Theoretically we could perform OOD detection with a single rollout instead of a batch during testing, but the detector and the corresponding bounds might be overly conservative. Hence, we use a batch during testing.

---

> > ### Author Response · Authors · 2021-08-28
> > **Continued Response to Reviewer ev1V**
> >
> > > The paper trains a policy with derandomized PAC Bayes. Would this affect the performance of the trained policy over the previous PAC-Bayes policies OR even the state-of-the-art methods?
> >
> > The derandomized PAC-Bayes bound allows us to perform OOD detection with a drastically smaller number of rollouts compared to the standard PAC-Bayes approaches (since the bound holds for a policy, not for a distribution on the policy space), thereby providing a significant advantage for OOD detection. For the manipulator, we obtained a bound of 0.1 with the derandomized PAC-Bayes compared to the standard PAC-Bayes bound of 0.07 reported in [42]; we do not have such a ready comparison available for the navigation example, but we expect the gap between the derandomized and standard PAC-Bayes to be small here as well.
> >
> > > How difficult is it to train as well? Also, how tight is the bound in practice?
> >
> > The details of the training are provided in Appendix A.4 and A.5 and the code for the training algorithms is also provided in the supplementary material. In practice we found that the PAC-Bayes guarantees are relatively tight. The test cost $C_\mathcal{D}(\pi)$ is around 0.02 (estimated with $500$ held-out environments) for the manipulator while the derandomized PAC-Bayes bound is 0.1. For the navigation example, the approximate $C_\mathcal{D}(\pi)$ (estimated with $50,000$ held-out environments) is $0.149$, whereas the PAC-Bayes bound is $0.222$.
> >
> > > In the appendix, the authors refer to different sources. For the researchers working on OOD detection, providing more background materials in the appendix can help in understanding the paper. For the researchers familiar with PAC-Bayes methods, it can help in not resorting to other papers.
> >
> > We have included an introduction the the PAC-Bayes framework in Appendix A.1.
> >
> > > An empirical validation of the proposed statistical guarantees can be meaningful, e.g. histogram analysis?
> >
> > We provide a numerical validation of Theorem 3 at the end of Appendix A.6.2. Here we show that with many samples, it is possible to violate the lower bound provided by the confidence interval analysis. However, since the guarantee holds with high probability (in this case 90\%) infrequent violations of the lower bound does not invalidate the theorem. We also show the 90th percentile line and that it is indeed below the lower bound as expected.

---

### Official Review · Reviewer_orAU · 2021-07-22

**Originality:** Very Good
**Technical Quality:** Very Good
**Clarity Of Presentation:** Excellent
**Impact:** 4

**Recommendation:**

Strong Accept: I recommend accepting the paper and will argue for my recommendation even if other reviewers hold a different opinion.

**Summary:**

This paper introduces a PAC-Bayes theory-based out-of-distribution detection (OOD) framework.
The paper proposes to train a policy whose related expected cost on the training distribution is bounded. After training the policy, OOD on the test environments can be performed with two different, proposed methods. The paper contributions are two-fold: guaranteed confidence bounds on OOD detection and sensitivity against environmental changes which matter (i.e. which changes the expected cost under the current policy).

**Issues:**

One major remark:

- I think the layout of the paper was changed, especially the space between equations and text. This should be addressed.

I have minor points that might be interesting to know for the reader:

- It would be interesting to know if the sample complexity of the training algorithms change due to optimizing the bound/"regularized cost"

- Theorem 1 requires the training algorithm to be deterministic, as stated in the Appendix. How does this influence the training procedure? Does this mean that usage of stochastic methods is restricted?

- What exactly is meant by transforming the training procedure into a supervised learning setting for the drone experiment?

**Reviewer Expertise:**

Poor: Limited knowledge of the area

**Strengths And Weaknesses:**

Strengths:

- provided methods have a bound guarantee which is proven mathematically
- paper is properly written, can be read fluently
- interesting and convincing experiments

Weaknesses:

- I don't see any main weaknesses
- Some more information regarding the policy training procedure would be nice (see Issues)

**Summary Of Recommendation:**

Even though I am not an expert in this field, I enjoyed reading the paper and learned a lot. The paper is clearly written and can be read fluently. The differences to prior work are made clear. The motivation and claims of the paper are proven via thorough mathematical derivations and experiments.

---

> ### Author Response · Authors · 2021-08-28
> **Response to Reviewer orAU**
>
> Thank you for your positive remarks and feedback. We address your comments below.
>
> > Layout of the paper was changed especially the space between equations and text
>
> We have fixed the layout and formatting of the paper. Additionally, to improve the flow of the paper we have moved some training-related technical details to the Appendix.  We have also added further explanations and details as requested by reviewers (highlighted in blue in the revision and discussed here in our responses to reviewers).
>
> > It would be interesting to know if the sample complexity of the training algorithms change due to optimizing the bound/"regularized cost"
>
> This is a very interesting topic to explore. Intuitively, we would expect the sample complexity to improve with the regularizer as it would prevent overfitting to the training dataset. Indeed, in alignment with our intuition, [43] observed better performance for policies trained with a PAC-Bayes regularizer for a grasping task. We have not deeply explored the sample complexity here since it is not the focus of the paper. Our primary use of the PAC-Bayes upper bound is to furnish a certificate of generalization which is then used for OOD detection.
>
> > Theorem 1 requires the training algorithm to be deterministic, as stated in the Appendix. How does this influence the training procedure? Does this mean that usage of stochastic methods is restricted?
>
> On fixing the random seed, stochastic methods are in fact deterministic as the same input will always produce the same output. Hence, practically the requirement of a deterministic algorithm is not detrimental and still permits the use of stochastic methods, given that we fix the random seed a priori. We have clarified this in the paper in lines 175-177.
>
> > Some more information regarding the policy training procedure would be nice. What exactly is meant by transforming the training procedure into a supervised learning setting for the drone experiment?
>
> We first recorded multiple rollouts of each primitive on the drone hardware. During training, which is performed entirely in simulation, the policy receives a depth map from its sensor for each environment in the training dataset. We leverage the simulation by generating a label for each depth map by recording the minimum distance to an obstacle achieved by each of the motion primitives (sampled uniformly from the set of recorded trajectories for that primitive) and passing the vector of distances through a softmax transformation. Note that we do not assume knowledge of the exact location of obstacles and record the closest distance as viewed by the robot's 120$^\circ$ field of view depth sensor. These depth maps and softmax labels can then be used for training the prior using supervised learning (where the input is the sensor measurement, the output is a softmax vector, and we use the cross entropy loss between the softmax output and label). We have included this explanation for improved clarity of our training procedure in Appendix A.6.2.

---

> > ### Comment · Reviewer_orAU · 2021-09-03
> > **Thanks for the response**
> >
> > I appreciate that the authors addressed my questions an concerns. I will stay with my recommendation.

---

### Official Review · Reviewer_PvzK · 2021-07-30

**Originality:** Good
**Technical Quality:** Very Good
**Clarity Of Presentation:** Very Good
**Impact:** 3

**Recommendation:**

Weak Accept: I recommend accepting the paper, but will not argue for my recommendation if the majority of other reviewers have a different opinion.

**Summary:**

This paper presents a method to detect OOD scenarios for robots. The method takes into account only changes that impact the robot’s performance, meaning that it neglects distribution shifts that do not affect task completion. This detection technique comes with guaranteed confidence bounds on performance, specifically on the expected cost of the policy. This bound is derived based on recently developed results on derandomized PAC-Bayes. The effectiveness of the method is shown on both simulated and real robotic experiments.


**Issues:**

To better inform future practitioners, you could give more intuition on the two methods you proposed, as in, when is better to use p-values vs concentration inequalities.


**Reviewer Expertise:**

Poor: Limited knowledge of the area

**Strengths And Weaknesses:**

### Strengths
The paper is very well written and structured. One can clearly see that the authors put a lot of care into defining everything thoroughly, and making sure that the readers do not have to go back and forth between sections. The experiments are also well designed and clearly organized. The results show an improvement over the baselines, which detect OOD even in cases with near-identical test expected cost (to the expected training cost), proving that the presented OOD detection is task-driven.
The task-driven formulation also helps when the robot’s sensor observations do not include possible disturbance factors as wind strength or direction.

### Weaknesses
As already pointed out by the authors in the conclusion, when using this pipeline one still incurs the risk to fail on the test environments, which is exactly what one wants to avoid, especially when dealing with autonomous vehicles.


**Summary Of Recommendation:**

This paper presents an elegant way to detect OOD environments and training a deterministic policy with a guaranteed bound on the expected cost. When the bound is violated, it indicates that the robotic system is not able to maintain the same performance - up a threshold - on the test set.
They propose two methods which both work better than the presented baselines. The experiments on the drone also show that the environment is not considered OOD until the wind speed is increased to 75% of the maximum disturbance. The main advantage of their method is two fold:
- not detecting OOD when the cost is unaffected by the changes, so in a way, not overestimating the proportion of OOD environments
- detecting OOD in scenarios where not task-driven methods would fail, as in the wind disturbance example

For this reason, for the clarity of the exposition and for the quality of the experiments, I would recommend acceptance.

---

> ### Author Response · Authors · 2021-08-28
> **Response to Reviewer PvzK**
>
> Thank you for your thoughtful and positive feedback. We address your comments below.
>
> > As already pointed out by the authors in the conclusion, when using this pipeline one still incurs the risk to fail on the test environments, which is exactly what one wants to avoid, especially when dealing with autonomous vehicles.
>
> Our approach does not necessarily require the system to fail; deterioration of the costs is sufficient to perform OOD detection. Indeed, this is demonstrated by our drone experiments where we are able to detect OOD environments without collisions since we rely on a cost signal that captures distance from the obstacles. Similarly, for the grasping example, one could perform OOD detection without dropping objects if one has access to a tactile sensor -- the cost signal would then correspond to the distance of the ratio of the normal and tangential forces from the boundary of the friction cone. Analogously, a self-driving car can use its distance to nearby obstacles/agents as a cost.
>
> However, we do agree with the reviewer that OOD detection without performing entire rollouts in test environments is a problem of great importance for safety-critical applications and we are currently working on addressing it. The following provides an insight into our approach for tackling this challenge:
>
> To achieve OOD detection with guaranteed false positive rate without performing entire rollouts, we are training a network that takes in a sensor measurement in a given environment and returns a prediction of the cost if the policy were to be deployed in this environment. We provide a PAC bound on the predicted cost; it is worth clarifying that the PAC bound is not on the true expected cost, but it is on the expected predicted cost for training environments. We leverage the insight that if the predicted cost for the test environments violate the PAC bound for the predicted cost on training environments, it indicates that the test environments are OOD. Finally, we note that we could perform this sort of OOD detection with a PAC bound on any scalar statistic, however, we use the predicted cost because it would allow us to embed task-relevance through the cost-predictor network. If a feature does not affect the cost of the policy, the cost predictor would ideally learn to discount it and focus on the task-relevant features.
>
> > To better inform future practitioners, you could give more intuition on the two methods you proposed, as in, when is better to use p-values vs concentration inequalities.
>
> In our experimental implementations, both our approaches --- p-value and confidence interval (CI) on the difference between the test and the train cost --- perform very similarly. However, the two approaches have complementary interpretations and cannot be derived from one another (clarified in the paper in line 49). The p-value bounds the probability of drawing test environments with average cost larger than the observed test cost, assuming that the robot is operating in a distribution that has the same expected cost as the training distribution (i.e., the null hypothesis holds). A small p-value thus indicates an unlikely outcome (under the null hypothesis). In contrast, the confidence interval (CI) approach directly estimates the difference between the expected cost on the test distribution and the expected cost on the training distribution (if this difference is positive, we can conclude with high probability that the robot is operating OOD in a task-relevant sense). Both approaches are thus of independent interest and utility.

---

> > ### Comment · Reviewer_PvzK · 2021-09-03
> > **Response to authors**
> >
> > I thank the authors for their reply and for the additional clarifications.
> > I find myself in line with Reviewer ev1V’s opinion.
> > They also pointed out a similar concern regarding the risk of failure and we received a common answer. This answer stressed the fact that it is not necessary for the agent to fail in order to detect OOD. Nevertheless, the first submitted video shows robot failure in case of strong wind disturbance. Since you need to perform entire rollouts, the probability of failure increases. This does not mean that failing is necessary to detect OOD, but that it could not be avoided - this is not ideal for safety-critical applications.
> > I also had some doubts regarding missing SOTA, but I am not an expert on this field, so I would refer to the other reviewers’ opinion on this matter.
> > Nevertheless, I appreciate the author’s effort to clarify all the concerns and I will keep my score as it is.

---

### Meta-Review · Area_Chair_7mYD · 2021-08-11

**Recommendation:** Accept (Poster)
**Confidence:** 4

**Metareview:**

The paper proposes a new technique for detecting when the robot is operating in an environment that is different from the one it trained in. This problem is known as out-of-distribution (OOD) detection, and it occurs often in robot learning, especially when new elements (such as wind) are introduced, or when a system is trained in simulation and transferred to the real world. Interestingly, the proposed method does not detect changes that do not affect the performance of the robot. The central idea behind this work is to train the robot with PAC-Bayes performance guarantees. When the robot operates in a given environment, it continuously measures its own performance and compares it against the predicted theoretical performance. When the the difference between the two is sufficiently large, based on p-values of hypothesis testing, then an OOD is detected. The proposed method is evaluated on 2 tasks in simulation and one real robotic task.
The reviewers agree that the technical contribution of this work is solid and novel. The paper is over all well written, although several technical details are not clear. It seems like the paper is written while assuming that the readers are all familiar with the PAC-Bayes literature and notations. Another issue raised by a reviewer is the utility of this approach in practice. It seems like OOD detection after a failure occurs is too late for sensitive robotic applications, such as self-driving cars. In these application, OOD should be tackled before a drop in the performance of the robot, contrary to the central idea of the proposed method here.

---

> ### Author Response · Authors · 2021-08-28
> **Response to Meta Reviewer**
>
> > The reviewers agree that the technical contribution of this work is solid and novel. The paper is over all well written, although several technical details are not clear.
>
> We thank the meta-reviewer for their favorable feedback. Following the reviewers' suggestions, we have included proof pathways before proving the theorems, elaborated the steps of the proofs more, added further intuition on the main results, and clarified technical details in the revised version of the paper.
>
> > It seems like the paper is written while assuming that the readers are all familiar with the PAC-Bayes literature and notations.
>
> We have revised the paper to include further details on the notation and added an introduction to the PAC-Bayes framework to the Appendix.
>
> > Another issue raised by a reviewer is the utility of this approach in practice. It seems like OOD detection after a failure occurs is too late for sensitive robotic applications, such as self-driving cars. In these application, OOD should be tackled before a drop in the performance of the robot, contrary to the central idea of the proposed method here.
>
> Our approach does not necessarily require the system to fail in order to perform OOD detection; deterioration of the costs is sufficient to perform OOD detection. Indeed, this is demonstrated by our drone experiments where we are able to detect OOD environments without collisions since we rely on a cost signal that captures distance from the obstacles. Similarly, for the grasping example, one could perform OOD detection without dropping objects if one has access to a tactile sensor -- the cost signal would then correspond to the distance of the ratio of the normal and tangential forces from the boundary of the friction cone. Analogously, a self-driving car can use its distance to nearby obstacles/agents as a cost.
>
> However, we do agree with the reviewer that OOD detection without performing entire rollouts in test environments is a problem of great importance for safety-critical applications and we are currently working on addressing it. The following provides an insight into our approach for tackling this challenge:
>
> To achieve OOD detection with guaranteed false positive rate without performing complete rollouts, we are training a network that takes in a sensor measurement in a given environment and returns a prediction of the cost if the policy were to be deployed in this environment. We provide a PAC bound on the predicted cost; it is worth clarifying that the PAC bound is not on the true expected cost, but it is on the expected predicted cost for training environments. We leverage the insight that if the predicted cost for the test environments violate the PAC bound for the predicted cost on training environments, it indicates that the test environments are OOD. Finally, we note that we could perform this sort of OOD detection with a PAC bound on any scalar statistic; however, we use the predicted cost because it would allow us to embed task-relevance through the cost-predictor network. If a feature does not affect the cost of the policy, the cost predictor would ideally learn to discount it and focus on the task-relevant features.

---

### Decision · Program_Chairs · 2021-09-13

**Decision:**

Accept (Poster)

**Comment:**

The paper proposes a new technique for detecting when the robot is operating in an environment that is different from the one it trained in. This problem is known as out-of-distribution (OOD) detection, and it occurs often in robot learning, especially when new elements (such as wind) are introduced, or when a system is trained in simulation and transferred to the real world. Interestingly, the proposed method does not detect changes that do not affect the performance of the robot. The central idea behind this work is to train the robot with PAC-Bayes performance guarantees. When the robot operates in a given environment, it continuously measures its own performance and compares it against the predicted theoretical performance. When the the difference between the two is sufficiently large, based on p-values of hypothesis testing, then an OOD is detected. The proposed method is evaluated on 2 tasks in simulation and one real robotic task.
The reviewers agree that the technical contribution of this work is solid and novel. The paper is over all well written, although several technical details are not clear. It seems like the paper is written while assuming that the readers are all familiar with the PAC-Bayes literature and notations. Another issue raised by a reviewer is the utility of this approach in practice. It seems like OOD detection after a failure occurs is too late for sensitive robotic applications, such as self-driving cars. In these application, OOD should be tackled before a drop in the performance of the robot, contrary to the central idea of the proposed method here.